# Structural, Electronic, and Optical Properties of Wurtzite $V_xAl_{1-x}N$ Alloys: A First-Principles Study

**Gene Elizabeth Escorcia-Salas** [1,2], **Diego Restrepo-Leal** [2,3], **Oscar Martinez-Castro** [4,5], **William López-Pérez** [4] **and José Sierra-Ortega** [2,*]

1 Grupo de Óptica e Informática, Departamento de Física, Universidad Popular del Cesar, Sede Hurtado, Valledupar 200001, Colombia; geneescorcias@unicesar.edu.co
2 Grupo de Investigación en Teoría de la Materia Condensada, Universidad del Magdalena, Santa Marta 470001, Colombia; diego.restrepoleal@campusucc.edu.co
3 Facultad de Ingeniería, Universidad Cooperativa de Colombia, Santa Marta 470001, Colombia
4 Departamento de Física, Universidad del Norte, Barranquilla 080001, Colombia; omartinez@uninorte.edu.co (O.M.-C.); wlopez@uninorte.edu.co (W.L.-P.)
5 Department of Basic Sciences, Universidad Simon Bolivar, Barranquilla 080001, Colombia
* Correspondence: jcsierra@unimagdalena.edu.co

**Abstract:** We present a comprehensive study on the structural, electronic, and optical properties of $V_xAl_{1-x}N$ ternary alloys using first-principles calculations. Our investigations employ the full-potential linearized augmented-plane-wave (FP-LAPW) method within the density functional theory (DFT) framework. The impact of varying vanadium composition ($x$ = 0, 0.25, 0.5, 0.75, 1) on the structural, electronic, and optical characteristics of wurtzite $V_xAl_{1-x}N$ alloys is examined in detail. Our findings reveal a distinct nonlinear relationship between the lattice constant, bulk modulus, and the concentration of vanadium ($x$) in the $V_xAl_{1-x}N$ alloys. An analysis of the electronic band structures and densities of states reveals a metallic behavior in the $V_xAl_{1-x}N$ alloys, primarily driven by the V-$d$ states near the Fermi energy. These results shed light on the electronic properties of the alloys, contributing to a deeper understanding of their potential for various applications. Furthermore, we calculate various optical properties, including the real and imaginary dielectric functions, refractive index, energy loss spectrum, and reflectivity. The obtained optical functions provide valuable insights into the optical behavior of the $V_xAl_{1-x}N$ alloys. The results contribute to the fundamental knowledge of these materials and their potential applications in various fields.

**Keywords:** DFT; AlN; V-doped AlN; electronic structure; optical properties

## 1. Introduction

Recently, there has been a growing interest in group-III nitrides due to their unique characteristics among III-V compounds. These nitrides are known for their high-temperature stability, short bond lengths, low compressibility, and high thermal conductivity. As a result, they have emerged as promising candidates for spintronic and optical devices [1]. With their wide band gaps, they are particularly well suited for electronic devices operating at high frequencies, high power levels, and elevated temperatures. Among the group-III nitrides, aluminum nitride (AlN) stands out as a prominent compound within the III-V family. It possesses a remarkable combination of physical, chemical, and mechanical properties, making it a highly desirable ceramic material. Notable features of AlN include its high thermal conductivity (320 W·m$^{-1}$·K$^{-1}$), excellent optical and dielectric properties, superior mechanical strength, strong corrosion resistance, thermal and chemical stability, and non-toxic nature [2]. These properties render AlN suitable for the design and production of electro-optical devices that operate under extreme conditions.

Currently, AlN finds applications in optical devices, surface-acoustic wave devices, high-power electronics, and high-temperature electronic devices. Its significance also extends to microelectronics, where it is widely recognized for its role as a thermal conductivity

and insulation material, particularly in electronic packaging [3–6]. Moreover, its wide band gap of approximately 6.2 eV, combined with a high refractive index of approximately 2.0 and a low absorption coefficient of less than $10^{-3}$, positions aluminum nitride as an attractive material for optical and optoelectronic applications [7]. Under ambient conditions, aluminum nitride adopts a wurtzite structure (space group *P6₃mc*) with lattice parameters *a* and *c* measuring 3.112 and 4.982 Å, respectively, [8].

In contrast, transition metals exhibit a wide range of properties, making them valuable for semiconductor applications and as dopant impurities for developing advanced materials with favorable electronic properties. Consequently, this subject has garnered significant attention and has been the focus of extensive research in recent years [9,10]. Some transition metal nitrides, such as VN, TiN, and NbN, have generated considerable interest due to their exceptional physical and mechanical properties. In the emerging field of spintronics, it is crucial to create semiconductors that possess ferromagnetically polarized carriers at room temperature (RT). This allows the spin and charge of the carriers to be coupled with an external magnetic field, enabling device control [11]. Researchers have pursued the development of dilute magnetic semiconductors (DMSs) by introducing minute amounts of transition metals (TM) into III-V, II-VI, and IV semiconductors [12–20]. DMS materials exhibit unique and intriguing properties, positioning them as potential candidates for spintronic and magnetoelectronic devices [21,22]. Notably, room-temperature ferromagnetism has primarily been observed in TM-doped systems, such as ZnO, GaN, and AlN [23–27]. Among these, AlN stands out due to its smallest lattice constant, suggesting the highest degree of sp-d hybridization between the host semiconductor and TM ions.

Recent theoretical and experimental studies have demonstrated that 3d transition metal (TM)-doped AlN, whether in bulk form or as nanosheets, exhibits magnetic ordering at room temperature [28–31]. Using density functional theory calculations, the electronic structure and magnetic properties of V-doped AlN were investigated [28]. The findings revealed that V dopants induce spin polarization upon substitutionally incorporating into AlN. Additionally, the formation energy suggests that V can be highly concentrated as a dopant, making it a promising candidate for AlN dilute magnetic semiconductors (DMSs). Due to their magnetic properties, DMS materials introduce an additional degree of complexity when incorporated into microcavities, making them interesting for studying light–matter interactions in confined systems [32]. By manipulating an external magnetic field, the energy levels and properties of excitons and magnetic ions can be adjusted in these systems [33]. This interaction can result in fascinating phenomena and new optical properties, providing opportunities to explore spin-related phenomena such as spin-dependent optical transitions, magneto-optical effects, and spin relaxation dynamics within the context of strong coupling [34]. In a strong coupling regime, the interaction between the cavity mode and excitons becomes significant, leading to the formation of lower-energy "exciton–polariton" states and higher-energy "photon" states. These hybrid states inherit properties from both photons and excitons, resulting in unique optical responses. The strong coupling regime is particularly intriguing due to its potential applications in quantum information processing, polariton lasers, and spintronics [35]. V-doped AlN cavities in a strong coupling regime offer a promising avenue for exploring fundamental physics, developing new types of device, and expanding the limits of light–matter interactions in the fields of photonics and optoelectronics [36].

Despite the existing body of research, there is a scarcity of studies focused on the properties of the $V_xAl_{1-x}N$ alloy. Furthermore, no systematic investigation has been conducted to explore the relationship between the electronic structure and optical properties of these compounds. In this study, we employ density functional theory calculations to examine the impact of V composition on the structural, electronic, and linear optical properties of wurtzite $V_xAl_{1-x}N$ alloys. Our analysis includes the real and imaginary components of the dielectric function, reflectivity spectra, and energy loss function, among others. The remainder of the paper is structured as follows: Section 2 outlines the computational details and theoretical methods employed in this study. Section 3 presents the results and discus-

sions on the investigated properties. Finally, in Section 4, we provide a concise conclusion summarizing the key findings of our study.

## 2. Computational Methods

To investigate the ternary $V_x Al_{1-x} N$ alloys, first-principles calculations based on density functional theory (DFT) were carried out. The alloys were modeled using ordered structures that are represented by periodically repeated cells. Figure 1 illustrates the crystal structures of the considered compositions in this study. For the V compositions with $x = 0.25, 0.5$, and 0.75, the resulting wurtzite cells exhibited two distinct values for the internal parameter, $u$. Each value corresponds to the separation, in units of $c$, between the nearest neighbors of Al and N (V and N). The optimization of $u$ was performed solely for the binary compounds AlN and VN in the wurtzite phase, and these optimized values were subsequently utilized for the other concentrations. To determine the total energy, the vanadium concentrations in the ternary alloy were varied across $x = 0, 0.25, 0.5, 0.75$, and 1. The Kohn–Sham equations were solved using the highly accurate full-potential linearized augmented-plane-wave plus local orbitals (FP-LAPW+lo) method, implemented in WIEN2K numerical code [37]. This approach allows for precise calculations of the electronic structure and total energy.

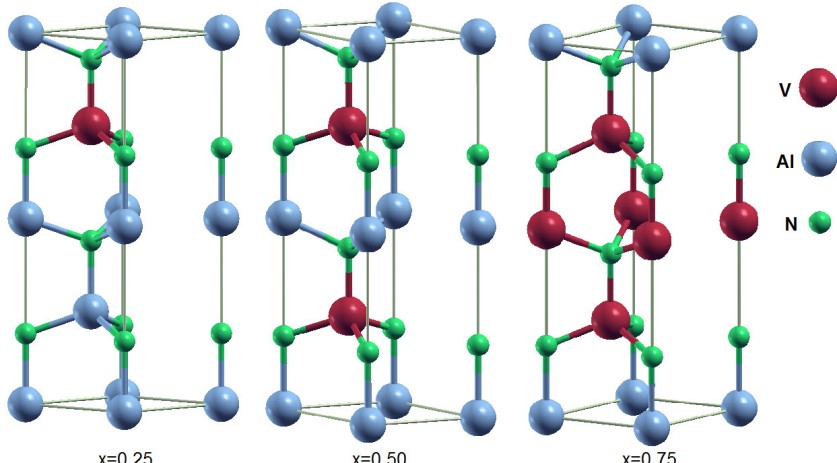

**Figure 1.** Supercells of $V_x Al_{1-x} N$ alloys in the wurtzite phases for V-compositions $x = 0.25$, 0.5, and 0.75. The red balls represent V atoms, the blue balls correspond to Al atoms, and the green balls represent N atoms.

It is important to note that the accuracy of band gap values obtained from density functional theory (DFT) calculations relies heavily on the choice of exchange–correlation (XC) potential, while the local density approximation (LDA) and generalized gradient approximation (GGA) [38] are commonly used for computing electronic and optical properties of solids; they are known to significantly underestimate energy band gaps. To address this limitation, the recent Tran–Blaha-modified Becke–Johnson (TB-mBJ) potential was utilized to calculate the electronic band structure [39]. A valence and core state energy of $-6.0$ Ry were considered in the calculations. To achieve convergence of energy values, plane-wave cutoffs were employed, with a maximum value of $K_{max} = 8/R_{mt}$, where $R_{mt}$ represents the smallest muffin-tin radius, and $G_{max} = 14 : Ry^{1/2}$ was used for charge density and potential expansion in the interstitial region. Wave functions within the muffin-tin spheres were expanded up to $l_{max} = 10$. The energy convergence criterion was set to $0.1 : mRy/Bohr$. For the V, Al, and N atoms, muffin-tin radii ($R_{mt}$) of 1.7, 1.9, and 1.6 Bohr, respectively, were selected. In the Brillouin zone, a mesh of 100 special k-points was employed for binary compounds, while 196 special k-points were used for ternary compounds with wurtzite phases, focusing on the irreducible wedge. Structural parameters were obtained by fitting the total energy versus volume data to Murnaghan's equation of state [40]. Subsequently,

electronic band structures were calculated at the equilibrium volume of $V_xAl_{1-x}N$ alloys for each V concentration using the TB-mBJ potential.

In order to accurately calculate the optical properties, it is essential to use a sufficiently large number of k-points in the Brillouin zone. This is due to the fact that the matrix element varies more rapidly within the Brillouin zone compared to the electronic energies themselves. Consequently, a higher number of k-points is required to precisely integrate this property, surpassing the requirements of a typical self-consistent field (SCF) calculation. Therefore, in this study, a dense k-point mesh of $20 \times 20 \times 20$ was employed for the calculations of optical parameters. By obtaining knowledge of the complex dielectric function, it becomes possible to derive the optical properties of the material. The imaginary part of the dielectric function was obtained from the electronic structure calculations, while the real part was derived using the Kramers–Kronig relation. From the values of the dielectric constant, it is then feasible to determine the refractive index and the optical reflectivity [41].

## 3. Results

### 3.1. Composition Dependence of Structural Properties

In order to have a description of the studied materials, we have optimized the atomic positions in the supercells by executing full structural relaxation to minimize the Hellmann–Feynman forces below $mRy/Bohr$ [42,43]. The resulting equilibrium volumes and total energies were utilized to evaluate various ground-state structural parameters for both the binary compounds AlN and VN, as well as the three permissible ternary compounds ($V_xAl_{1-x}N$ with $x = 0.25$, 0.50, and 0.75) in the wurtzite phases. These parameters include the lattice parameter ($a$), equilibrium volume ($V_0$), bulk modulus ($B_0$), and cohesive energy ($E_{coh}$). To determine these parameters, we calculated the total energy as a function of volume and subsequently fitted the results to Murnaghan's equation of state [40]. The calculated values for the structural parameters of the wurtzite $V_xAl_{1-x}N$ alloys at different compositions ($x$) are summarized in Table 1, alongside the available experimental data for comparison.

**Table 1.** Calculated lattice parameter ($a$), $c/a$ ratio, bulk modulus ($B_0$), and equilibrium volume (per unit formula) ($V_0$) and cohesive energy $E_{coh}$ for $V_xAl_{1-x}N$ alloys in wurtzite structures.

| $x$ | $a$(Å) | $c/a$ | $B_0$ (GPa) | $V_0(^3)$ | $E_{coh}$ (eV) |
|---|---|---|---|---|---|
| 0 | 3.139 | 1.602 | 196 | 21.436 | −14.839 |
| | 3.112 [a] | 1.601 [a] | | | |
| | 3.123 [b] | | 192 [b] | | |
| 0.25 | 3.126 | 1.66 | 202 | 21.939 | −15.403 |
| 0.50 | 3.110 | 1.71 | 201.6 | 22.232 | −16.051 |
| 0.75 | 3.105 | 1.70 | 219.3 | 22.071 | −17.101 |
| 1 | 3.094 | 1.64 | 227.08 | 21.898 | −18.167 |
| | 3.10 [c] | | 209 [c] | | |

[a] Ref [8], [b] Ref [44], [c] Ref [45].

Under ambient conditions, pure AlN exhibits a wurtzite crystal structure characterized by a lattice constant of 3.112 Å [8]. In our calculations using the PBE exchange–correlation functional, the determined lattice constant ($a$) for wurtzite AlN is found to be larger than the experimental value by 0.86%. Similarly, the parameter $c$ is overestimated by 0.96%. These results are consistent with the well-known tendency of the GGA approach to overestimate lattice constants. Furthermore, the percentage difference in lattice constants between VN and AlN is approximately 1.44%. This observation suggests that combinations of VN and AlN could be advantageous in the growth of heterostructures involving AlN/VN or in the synthesis of V-doped AlN alloys. Such combinations hold potential for various applications and can be explored in the design and fabrication of novel materials with tailored properties.

Figure [2]a illustrates a decrease in the lattice constant as the vanadium composition increases. The calculated lattice constants of the $V_x Al_{1-x} N$ alloys at various vanadium compositions exhibit a slightly nonlinear dependence on the concentration of vanadium, denoted by x. The variation in the equilibrium lattice constant, as calculated, deviates from Vegard's law [46], showing an upward bowing parameter of approximately $-0.01226$ Å. This deviation can be attributed to the subtle differences in the lattice constants of the binary compounds. Such a small negative deviation from Vegard's law is commonly observed in this type of compounds and aligns with the compositional variation of lattice parameters observed in conventional III-V group alloys. Recent theoretical investigations [47] have also indicated the invalidity of Vegard's law for III-V nitrides. In Figure [2]b, the bulk modulus behavior as a function of vanadium composition in the $V_x Al_{1-x} N$ alloy is depicted, along with a comparison to the linear composition dependence (LCD). The calculated bulk modulus increases with higher vanadium concentrations, indicating reduced compressibility as vanadium content increases. A minor deviation from the LCD can be observed, characterized by a downward bowing parameter of approximately 23.5.

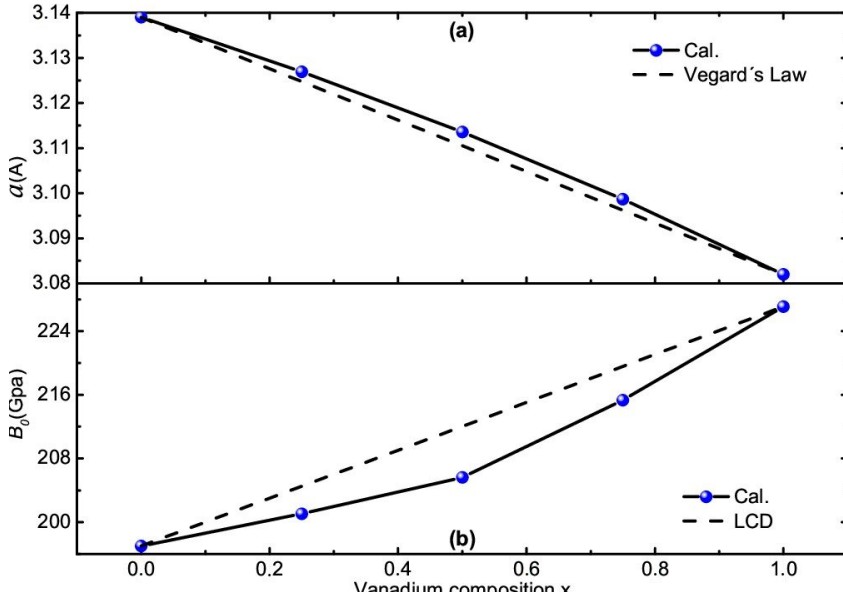

**Figure 2.** (**a**) Composition dependence of the calculated lattice constant (*a*) for wurtzite $V_x Al_{1-x} N$ alloys. The calculations (spheres) are compared with Vegard's linear tendency (dashed line). (**b**) Composition dependence of the bulk modulus of wurtzite $V_x Al_{1-x} N$ alloys. The calculations (spheres) are compared with the linear composition dependence (dashed line).

### 3.2. Electronic Structure

To study the electronic structure of $V_x Al_{1-x} N$ alloys in the wurtzite phase, the band structure and the density of states (DOSs) in the equilibrium volume were calculated using the TB-mBJ potential. Our mBJ results confirm that AlN is a direct band gap semiconductor with $\Gamma \to \Gamma$ energy of 5.385 eV, which is in agreement with the GW and mBJ results reported by other authors. Vanadium nitride shows a metallic behavior, as shown in Figure [3]. For intermediate concentrations, $x = 0.25, 0.50,$ and $0.75$, all structures exhibit a metallic behavior. This indicates that the vanadium atoms originate a metallization of the $V_x Al_{1-x} N$ alloys, which is important from an experimental point of view. To obtain a deeper insight into the electronic structure, the total and partial densities of states (TDOS and PDOS) for $V_x Al_{1-x} N$ alloys in the wurtzite phase were also calculated. Figure [4] shows the calculated TDOS and PDOS for wurtzite $V_x Al_{1-x} N$ alloys.

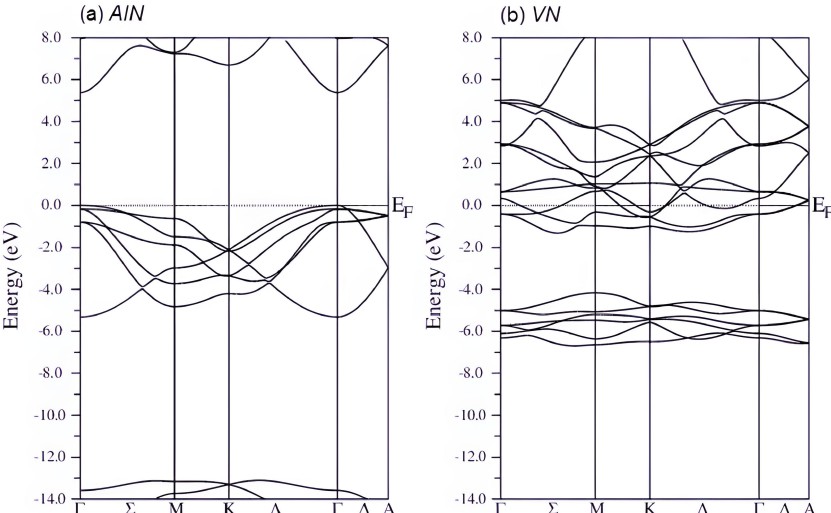

**Figure 3.** Band structure for (**a**) AlN and (**b**) VN in a wurtzite configuration, at the ground state. The valence-band maximum is the zero energy level.

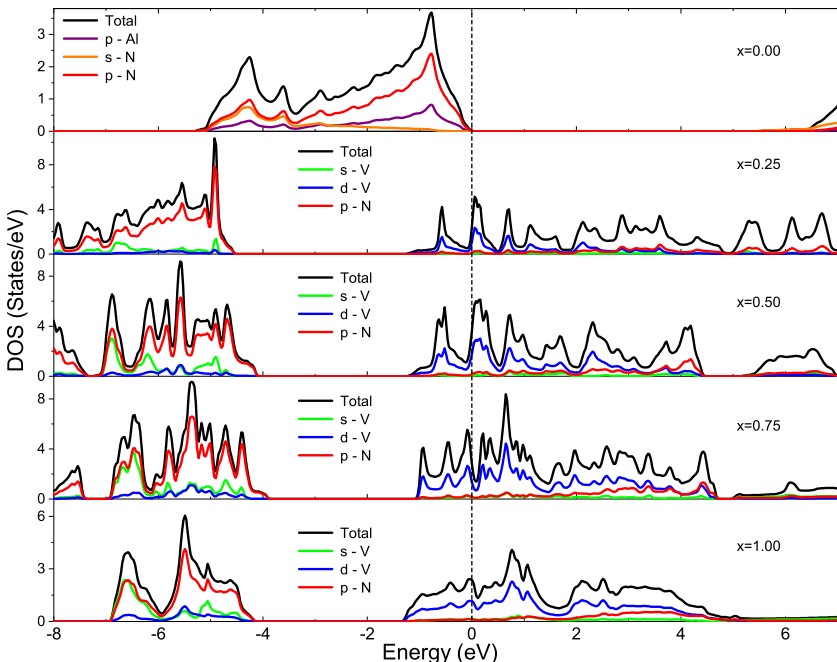

**Figure 4.** Total and partial densities of states in equilibrium volume of wurtzite $V_xAl_{1-x}N$ alloys. The dotted vertical line indicates the zero energy level.

From this figure, it can be noted that the valence band is mainly dominated by N-p states with some contributions from V-d. It is evident that the metallic character of the alloy for all concentrations is determined mainly by the contribution of V-d states close to the Fermi energy. Therefore, the incorporation to extra electrons from the V atom move the Fermi level toward the conduction band. This phenomena effect eliminates the band gap energy of the aluminum nitride, which introduces a metallic character to the $V_xAl_{1-x}N$ alloy. The contribution of the V-d states around the Fermi level increases as the vanadium concentration increments and also the presence of these electrons increases above the Fermi level. The contribution of the N-p states to the conduction band is reduced with the increase in the vanadium concentration. The states of N-p and V-d are at the maximum of the valence band and, therefore, could be responsible for the electric conduction in the ternary material. Electronic structures show a qualitative behavior similar to that of other systems, as $Ni_xAl_{1-x}N$ [48], $Ag_xAl_{1-x}N$ [49], $V_xGa_{1-x}N$ [50], and $CuGaSe_2$ [51].

### 3.3. Optical Properties

The optical properties of solids play a crucial role in the analysis and design of optoelectronic devices, including light sources and detectors. These properties arise from the electronic excitations within crystals when they are subjected to electromagnetic radiation. A comprehensive description of the optical properties is often accomplished using the complex dielectric function, denoted as $\varepsilon(\omega) = \varepsilon_1(\omega) + i\varepsilon_2(\omega)$. Here, $\varepsilon_1(\omega)$ and $\varepsilon_2(\omega)$ represent the dispersive and absorptive components of the dielectric function, respectively. Typically, the dielectric function consists of two main contributions: intraband and interband transitions. The intraband transitions are primarily significant for metals. On the other hand, the interband transitions can be categorized into direct and indirect transitions. Indirect interband transitions involve the scattering of phonons, but their contribution to $\varepsilon(\omega)$ is usually small compared to direct transitions. Consequently, we have neglected the indirect transitions in our calculations, focusing on the dominant direct transitions.

To gain insights into the optical properties of wurtzite $V_x Al_{1-x}N$ alloys and provide valuable guidance for the design of optical devices utilizing these nitrides, specifically, we focused on calculating the dielectric function, refractive index, reflectivity coefficients, and energy-loss spectrum of V-doped AlN systems. The results we present here pertain to two components of the polarization vector, which correspond to the average of the spectra for polarizations along the $x$ and $y$ directions, with the polarization vector aligned parallel to the $z$ axis. In order to determine the electronic band structures, we employed the TB-mBJ potential and performed calculations at the equilibrium volume of the $V_x Al_{1-x}N$ alloys for each specific vanadium concentration. These calculations allow us to explore the optical properties of the alloys and provide valuable information for the material design of optical devices.

### 3.3.1. Dielectric Function

In the linear response range, the macroscopic optical response functions of solids are often described by the complex dielectric function. Knowing the real $\varepsilon_1(\omega)$ and imaginary $\varepsilon_2(\omega)$ parts of the dielectric function, one can calculate several important optical characteristics (absorption, reflectivity coefficients, complex refractive index, etc.).

Figure 5 displays the calculated real (left side) and imaginary (right side) parts of the dielectric function $\varepsilon(\omega)$ as a function of photon energy up to 25 eV for the wurtzite phase of the $V_x Al_{1-x}N$ compound, where $x$ takes values of 0.0, 0.25, 0.50, 0.75, and 1.0. We have separated the dielectric function into two components: $\varepsilon_{100}(\omega)$ and $\varepsilon_{001}(\omega)$. These components represent the average spectra for polarizations along the $x$ and $y$ directions, respectively, with the polarization vector parallel to the $z$ direction ($z$ axis). Upon comparing the curves of $\varepsilon_{100}(\omega)$ and $\varepsilon_{001}(\omega)$, it is evident that an anisotropy exists between these components across the crystal structure for all cases ($x$ = 0.0, 0.25, 0.50, 0.75, and 1.0). In the case of $x$ = 0.0, corresponding to pure AlN, the $\varepsilon_2(\omega)$ curves in Figure 5 indicate that the threshold energy occurs at 5.385 eV.

This is the first critical point of the dielectric function, which corresponds to the $\Gamma_v - \Gamma_c$ splitting of wurtzite AlN, as shown in Figure 3, and gives the threshold for direct optical transitions between the highest valence and the lowest conduction bands, also known as fundamental absorption edge. Beyond these points, the curve increases rapidly, which is due to the fact that the number of points contributing towards $\varepsilon_2(\omega)$ increases abruptly. When the energy increases, the absorptive part of the dielectric function exhibits two major peaks located at about 9.25 and 11.9 for $\varepsilon_{100}(\omega)$, as well as 8 and 10.8 eV for $\varepsilon_{001}(\omega)$, of the wurtzite AlN. Thus, the first spectral peak shows electron transitions from valence band maximum to conduction band minimum. The first peak is dominated by transition mainly from N-p states in the highest valence band to Al-p states in the lowest conduction band, while the second peak corresponds to the transitions of N-p electrons into Al-s conduction bands. Some weak peaks can be observed between these two peaks that are related to weak resonances in the structure. For the concentration $x = 0.25$, $\varepsilon_{001}(\omega)$ exhibits a relevant peak at 7.7 eV and an energy threshold at 0.5 eV, while a rapid increase occurs at 2.5 eV for

$\varepsilon_{100}(\omega)$ due to anisotropy effects. For the concentrations $x = 0.5$, 0.75, and 1, the curves show some outstanding peaks in an energy range of 5 to 9 eV, exhibiting greater intensity in the $\varepsilon_{001}(\omega)$ curve. In addition, a rapid increase occurs at 4 eV for both curves, and small values of the imaginary dielectric function are presented above 15 eV. This behavior is related to the intraband transitions in the valence band, which can realize the N-p and V-d electrons mainly, as observed in the density of states. Some weak peaks can be observed in the higher energies, which are related to weak resonances in the structure. The curve $\varepsilon_1(\omega)$ shows a minimum energy of 15 eV for AlN. The real part value in the zero energy limit is 3.85 eV. For $x = 0.25$, the real dielectric function is very different from each direction, indicating a high degree of anisotropy. At the other concentrations, a less drastic difference can be observed, and the curves have a minimum (maximum) below 2 eV (1 eV).

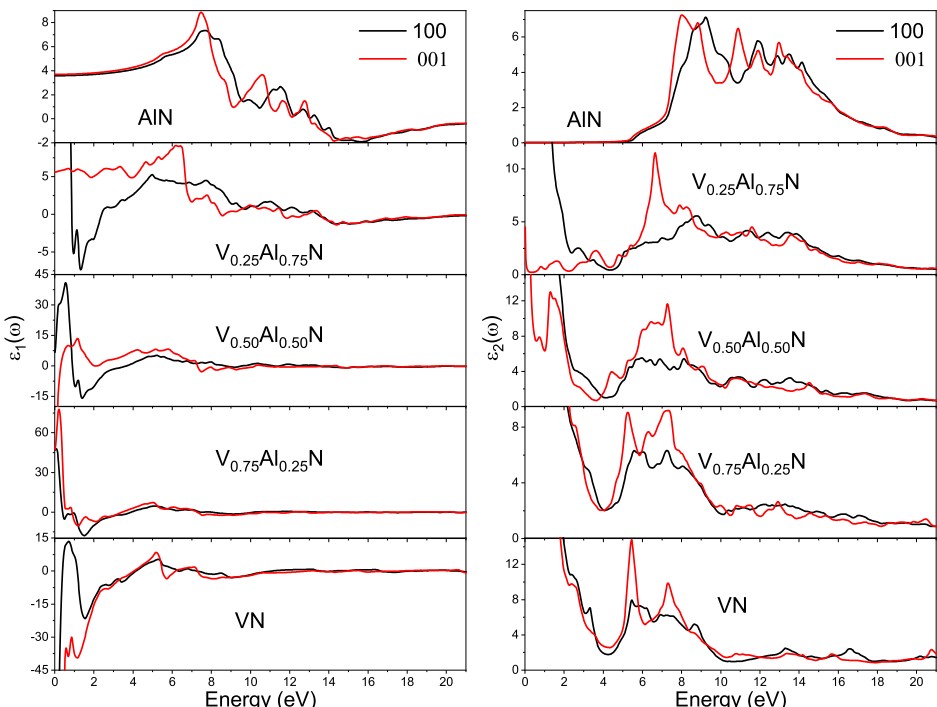

**Figure 5.** Real (**left side**) and imaginary (**right side**) of the dielectric function of wurtzite $V_x Al_{1-x} N$ alloys.

### 3.3.2. Refractive Index and Extinction Coefficient

The calculated refractive index along xx- and yy-direction are shown on left side in Figure 6. A slight variation in the refractive index as a function of the direction can be observed for concentrations $x = 0$, 0.75, and 1. At compositions $x = 0.25$ and 0.5, the refractive index shows a considerable difference as a function of the direction at low energies. The static refractive index in the zero frequency limit is about 1.93, 13, and 12.8 for $x = 0$, 0.75, and 1, respectively. For $x = 0.25$, the refractive index at zero frequency is about 2.44 and 7 in the (001) and (100) directions, respectively. For $x = 0.25$, at zero frequency, the refractive index value is about 5.2 and 8 in the (001) and (100) directions, respectively. The refractive index presents a maximum value of 3.07 at 7.58 eV for AlN; while for the remaining concentrations, the maximum value of the refractive index occurs in very small energies except for the direction (100) at $x = 0.25$, the value of which is 3.25 at 6.51 eV. The extinction coefficient is plotted on the right side in Figure 6. Since the extinction coefficient is the imaginary part of the refractive index, they exhibit a similar behavior as a function of the direction. They also present a very similar behavior in the shape of the curves at concentrations $x = 0.25$, 0.5, 0.75, and 1. However, the AlN has no extinction coefficient below 4.85 eV.

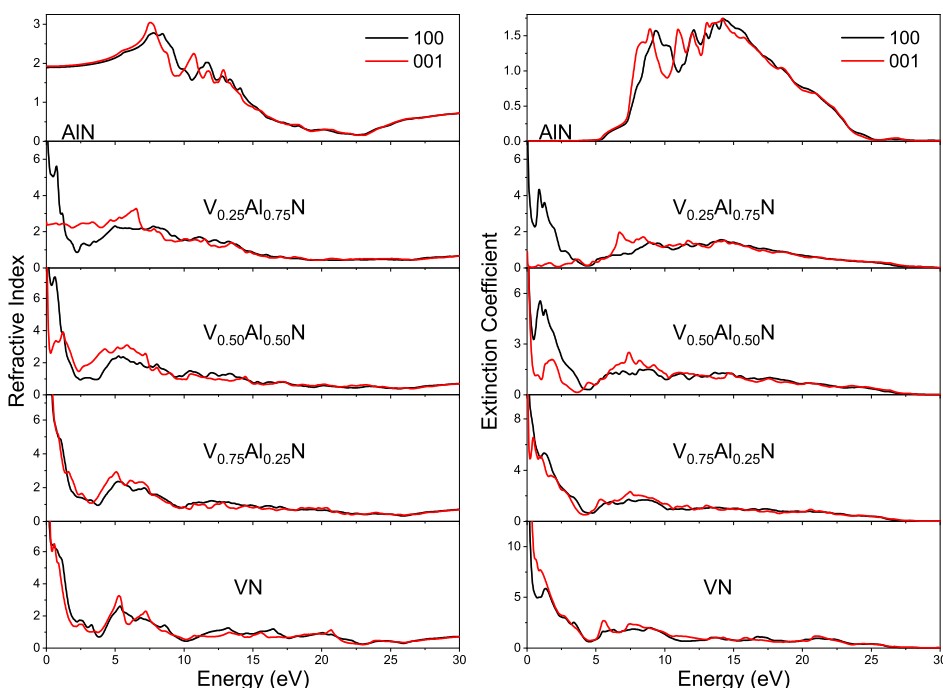

**Figure 6.** The calculated refractive index $n(\omega)$ (**left side**) and extinction coefficient (**right side**) for wurtzite $V_x Al_{1-x} N$ alloys.

### 3.3.3. Energy Loss Function and Reflective Spectra

The electron energy loss function is an important factor describing the energy loss of a fast electron traversing in a material. The peaks that can be seen in the loss function spectra are related to the resonance frequency called plasma oscillation frequency. The energy loss can be obtained in terms of dielectric function. The loss function curves of $V_x Al_{1-x} N$ alloys are plotted in Figure 7 (left side). It can be seen from Figure 7 (left side) that the height of the peaks is dependent on the direction. For AlN, the loss function peaks are located at 23.39 eV; however, the peak value along (001) is higher than that of (100). Thus, the characteristic of plasma oscillation along (001) is better than that of (100) for AlN. The $V_{0.25} Al_{0.75} N$ alloys show a peak value higher along (100) than in the direction (001), located in 25.76 and 26.60 eV, respectively. For $x = 0.5$, the maximum value of loss function is found along (001); and the peaks are located at 26.62 and 25.61 eV along (001) and (100), respectively. For $x = 0.75$, the best plasma oscillation behavior is along (100) with a maximum value at 25.95 eV. In the VN compound, a behavior similar to that of the concentration $x = 0.75$ is presented. In Figure 7 (right side), the reflectivity spectra of the $V_x Al_{1-x} N$ alloys as a function of the energy are shown. The maximum value for AlN is located at 25.55 eV along (001) and shifts to 0 eV for VN due to the intraband decrease in the valence band. The spectrum has a similar trend in shift with an increase in V-concentration. For high energy values, the reflectivity reaches a very small stable value.

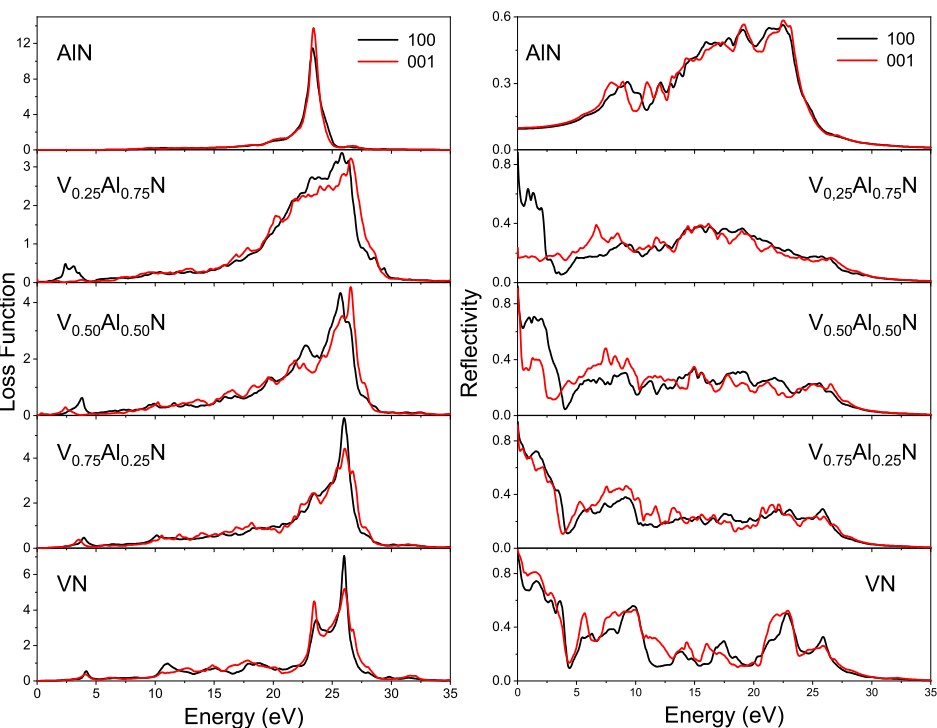

**Figure 7.** The calculated loss function $L(\omega)$ (**left side**) and optical reflectivity spectrum $R(\omega)$ (**right side**) for wurtzite $V_x Al_{1-x} N$ alloys.

## 4. Conclusions

In summary, our comprehensive investigation of the structural, electronic, and optical properties of wurtzite $V_x Al_{1-x} N$ alloys using density functional theory has yielded significant insights into the fundamental nature of these materials. From a structural perspective, we observed a consistent decrease in the equilibrium lattice constant as the vanadium concentration increased, indicating a clear correlation between composition and lattice parameters. Additionally, the alloys exhibited a weak negative deviation from Vegard's law, highlighting the influence of alloying on the lattice structure. The increasing trend in bulk modulus with vanadium composition suggests enhanced mechanical rigidity, making these alloys promising candidates for applications requiring high-temperature and high-power operation, as well as for robust coatings. The investigation of the electronic properties revealed intriguing features of the alloy system; while pure AlN displayed the characteristics of a direct band gap semiconductor, the introduction of vanadium induced a shift toward metallic behavior in the alloy composition. This shift was accompanied by an increase in the contribution of vanadium-d states around the Fermi level and a decrease in the contribution of nitrogen-p states to the conduction band. These electronic findings underline the significance of alloying VN with AlN and highlight the potential for engineering the electronic properties of these materials for electronic device applications. Furthermore, our study provided a comprehensive analysis of the optical properties of $V_x Al_{1-x} N$ alloys. The evaluation of various optical parameters, including the dielectric function, refractive index, reflectivity coefficients, and energy-loss spectrum, elucidated the optical behavior of the alloys. Notably, we observed significant anisotropy between two components of the polarization vector throughout the crystal structure, underscoring the importance of considering the polarization direction in the design and optimization of optoelectronic devices based on these alloys. Overall, our findings contribute to a deeper understanding of the structural, electronic, and optical aspects of wurtzite $V_x Al_{1-x} N$ alloys. These insights provide crucial guidance for the development of doped alloys and pave the way for their utilization in electronic and optoelectronic technologies, where tailored properties are essential for achieving enhanced performance and functionality.

**Author Contributions:** G.E.E.-S., conceptualization, methodology, formal analysis, and writing—original draft preparation, review, and editing; D.R.-L. and O.M.-C., formal analysis and writing; W.L.-P. and J.S.-O., methodology, formal analysis, and writing. All authors have read and agreed to the published version of the manuscript.

**Funding:** This work was financed by the Universidad del Magdalena and Universidad Popular del Cesar through Vicerrectoría de Investigación, and the Universidad del Norte through Departamento de Física.

**Data Availability Statement:** No new data were created or analyzed in this study. Data sharing is not applicable to this article.

**Conflicts of Interest:** The authors declare no conflict of interest.

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
