# Peer review of "Structural, Electronic, and Optical Properties of Wurtzite VxAl1−xN Alloys: A First-Principles Study"

_condensedmatter, doi:10.3390/condmat8030061_

Round 1
Reviewer 1 Report
This paper by Salas et al. covers an interesting simulation study of tertiary nitride semiconductors. The paper is well written, with good use of English, and the extension of DFT methods into the key optical properties of various V-Al-N alloys is nicely presented. However, before publication, I would expect to see the following points address in a revised manuscript.
1. The stated interest in these materials (Introduction) is the DMS regime, where the level of transition metal substitution is <<1. Yet the manuscript only deals with specific cases of x = 0.25, 0.5, 0.75. Whilst the change from semiconducting to conducting properties with V addition is nicely illustrated, the study would be greatly improved, in my opinion, but the addition of results in the small x regime. This would ideally show the point at which a changeover in the properties occurs as x is varied?
2. I am slightly surprised to see that the authors state that "no new data were created" (line 322). Subject to the guidelines offered by the journal and publisher, I'd expect that numerical outputs from simulations, as used in the figures, should be archived in the same way that experimental data would be? I recognise that the methodology of DFT is well described in Section 2, and this would help with future reproducibility and/or extensions of the study.
Reviewer 2 Report
AlN and VN are among the most studied nitrides due to their potential applications in optical and spintronic devices, respectively. Both cystallize in a wurtzite structure with similar lattice constants. This has led experimentalists to realize V-doped AlN materials. Authors present a high-level ab-initio study of the properties of the two perfect compounds as well as three intermediate compounds. They studied, for the first time, the variation of the structural, electronic and optical properties as a function of the V and Al concentrations. Their study is novel. Results and conclusions are well-supported by the data. Figures/tables/references are adequate. The level of English is high. Motivation is clear and overall the manuscript is of high level. Thus I recommend its acceptance with no further revision.
Author Response
We thank you very much for the analysis of our manuscript
Round 2
Reviewer 1 Report
I am happy with the recognition and response from the authors about my concerns re: DMS regime. I'm sure this will be a focus of further studies. For the sake of reader clarity, I would suggest perhaps adding a short comment about this to the Conclusions section?